

# When and where to hatch? Red-eyed treefrog embryos use light cues in two contexts

Brandon A. Güell[1,2] and Karen M. Warkentin[2,3]

[1] University of California, San Diego, CA, United States of America
[2] Department of Biology, Boston University, Boston, MA, United States of America
[3] Smithsonian Tropical Research Institute, Panama City, Panama

## ABSTRACT

Hatching timing is under strong selection and environmentally cued in many species. Embryos use multiple sensory modalities to inform hatching timing and many have spontaneous hatching patterns adaptively synchronized to natural cycles. Embryos can also adaptively shift their hatching timing in response to environmental cues indicating immediate threats or opportunities. Such cued shifts in hatching are widespread among amphibians; however, we know little about what, if anything, regulates their spontaneous hatching. Moreover, in addition to selection on hatching timing, embryos may experience benefits or suffer costs due to the spatial orientation of hatching. Amphibian eggs generally lack internal constraints on hatching direction but embryos might, nonetheless, use external cues to inform hatching orientation. The terrestrial embryos of red-eyed treefrogs, *Agalychnis callidryas*, hatch rapidly and prematurely in response to vibrational cues in egg-predator attacks and hypoxia if flooded. Here we examined *A. callidryas'* use of light cues in hatching timing and orientation. To assess patterns of spontaneous hatching and the role of light cues in their diel timing, we recorded hatching times for siblings distributed across three light environments: continuous light, continuous dark, and a 12L:12D photoperiod. Under a natural photoperiod, embryos showed a clear diel pattern of synchronous hatching shortly after nightfall. Hatching was desynchronized in both continuous light and continuous darkness. It was also delayed by continuous light, but not accelerated by continuous dark, suggesting the onset of dark serves as a hatching cue. We examined hatching orientation and light as a potential directional cue for flooded embryos. Embryos flooded in their clutches almost always hatched toward open water, whereas individual eggs flooded in glass cups often failed to do so, suggesting the natural context provides a directional cue. To test if flooded embryos orient hatching toward light, we placed individual eggs in tubes with one end illuminated and the other dark, then flooded them and recorded hatching direction. Most embryos hatched toward the light, suggesting they use light as a directional cue. Our results support that *A. callidryas* embryos use light cues to inform both when and where to hatch. Both the spatial orientation of hatching and the timing of spontaneous hatching may affect fitness and be informed by cues in a broader range of species than is currently appreciated.

Corresponding author
Brandon A. Güell, bguell@bu.edu

## INTRODUCTION

Maternally provided structures such as egg capsules and shells provide embryos with protection, yet limit their interaction with the surrounding environment. Thus, at hatching animals enter new physical environments, with new dangers and resources. Many biotic and abiotic factors differentially affect eggs and hatchlings, creating selective tradeoffs that determine the optimal time or developmental stage at which to hatch, often in a context-dependent manner (*Christy, 2011*; *Doody, 2011*; *Korwin-Kossakowski, 2012*; *Warkentin, 2011a*). A substantial body of research has documented adaptive shifts in hatching timing in response to cues indicating immediate, direct threats to embryo survival (e.g., hypoxia (*Petranka, Just & Crawford, 1982*; *Warkentin, 2007*), predation, dehydration, and pathogens (reviewed in *Warkentin, 2011a*)). However, embryos may also use environmental cues to inform their hatching timing or process in the absence of immediate threats. For instance, embryos may time hatching to occur at favorable points during natural photoperiod, tidal, and lunar cycles (*Christy, 2011*). Embryos may also orient hatching in relation to the structure of their egg capsule or its immediate surroundings in ways that improve their hatching process or post-hatching fate (*Korwin-Kossakowski, 2012*; *Oppenheim, 1972*).

Hatching during low-risk periods of natural cycles has been widely documented among crustaceans and fishes. For example, many tropical brachyuran crabs and reef fishes release their larvae during high-amplitude, nocturnal high tides, facilitating larval transport away from high-predation locations (*Christy, 2003*; *Johannes, 1978*; *Morgan & Christy, 1995*; *Robertson, 1991*). This regulation of hatching timing in response to abiotic cycles in fish and crustacean embryos is hypothesized to be an adaptive strategy which serves as an antipredator defense mechanism (*Christy, 2003*; *McAlary & McFarland, 1993*; *Morgan & Christy, 1995*). It can also serve, for parasites, to increase their chances of finding a host (e.g., monogenean flatworms, *Whittington & Kearn, 2011*). However, such hatching responses are complex and may be endogenously or exogenously regulated, or both (see *DeCoursey, 1983*). For instance, in different species of brachyuran crabs, hatching timing may be controlled by either the mother or embryos' endogenous rhythms, which are set and may be altered by environmental cues and cycles (reviewed in *Christy, 2011*). One common and widespread environmental factor which entrains endogenous hatching rhythms are light-dark cycles (*Asoh & Yoshikawa, 2002*; *Brännäs, 1987*; *Brüning, Hölker & Wolter, 2011*; *DeCoursey, 1983*; *McAlary & McFarland, 1993*; *Salmon, Seiple & Morgan, 1986*).

Environmentally cued shifts in hatching timing are widespread in amphibians (*Warkentin, 2011b*). However, hatching rhythms that may concentrate hatching at favorable times in natural cycles have received little attention, despite the likelihood of diel variation in risks to eggs and hatchlings. Such cycles of risk may be particularly relevant for species that lay terrestrial eggs above water, into which tadpoles fall upon hatching. Embryos presumably have no direct information about current conditions in their larval habitat but, for instance, a history of tadpole predation by visual predators, such as fishes or odonates, could select for hatching under the cover of darkness. Here, we use one such species, the

red-eyed treefrog *Agalychnis callidryas* (Cope 1862), to assess the effect of light-dark cycles on the timing of hatching.

The embryos of *A. callidryas* develop in gelatinous egg masses on leaves above ponds, in lowland wet forests from Mexico to Colombia. Their hatching is a rapid process, involving enzyme release and exit behavior, that embryos may perform at any point across a broad developmental period; the actual hatching timing of individual embryos depends on their behavioral decision (*Cohen, Seid & Warkentin, 2016*; *Warkentin et al., 2017*). Embryos exposed to predator attack, flooding, fungal infection, or dehydration can hatch rapidly to escape, starting as early as 3 or 4 days of age (*Salica, Vonesh & Warkentin, 2017*; *Warkentin, 1995*; *Warkentin, 2000b*; *Warkentin, 2007*; *Warkentin et al., 2017*; *Warkentin, Currie & Rehner, 2001*). In contrast, undisturbed embryos often remain in the egg for 5–7 d (*Gomez-Mestre & Warkentin, 2007*; *Hite et al., 2018*; *Warkentin, 1995*; *Warkentin, 2000b*). Undisturbed egg clutches, and those facing the chronic threat of fungus or drying, tend to hatch gradually, often taking over 24 h for all eggs to hatch, while embryos facing immediate threats in attacked clutches may hatch synchronously, within minutes of each other (*Salica, Vonesh & Warkentin, 2017*; *Warkentin, 1995*; *Warkentin, 2000b*; *Warkentin, Currie & Rehner, 2001*). Aquatic predator communities in *A. callidryas* breeding ponds vary substantially; however, hatchlings often face visually hunting insects or fishes (*Touchon & Vonesh, 2016*). Moreover, in clutches monitored at ponds, the majority of the embryos hatch during the night (*Gomez-Mestre & Warkentin, 2007*; *Warkentin, 1995*). To evaluate the potential role of direct light cues and endogenous rhythms in the hatching timing of undisturbed *A. callidryas* embryos, we manipulated embryos' light environments. We raised embryos in one of three light treatments: continuous light, continuous darkness, and a 12L:12D photoperiod (Fig. 1A). It is possible that embryos have an endogenous hatching rhythm; if so, we would expect hatching to be temporally patterned (i.e., concentrated during a particular diel period) even in the absence of photoperiod cues, and thus similar across treatments. This could explain nocturnal hatching in nature (*Gomez-Mestre & Warkentin, 2007*; *Warkentin, 1995*); however, based on prior observations of varying diel hatching patterns under artificial lights in the lab, we hypothesize that hatching is cued by darkness or its onset, or inhibited by light. Thus, we expect differences in hatching timing across photoperiod treatments. Specifically, if darkness *per* se stimulates hatching, we expect embryos in continuous darkness to hatch earliest. If light inhibits hatching we expect those in continuous light to hatch latest. Finally, if the change from light to dark cues hatching we expect the greatest hatching synchrony in the photoperiod treatment, with a peak at the onset of darkness.

In addition to better and worse times to hatch, there may be better and worse spatial orientations for hatching. The impact of fetal orientation on the emergence process is well-known for mammalian birth, where breech positions are associated with increased complications (*Hannah et al., 2000*). However, embryo position during hatching is also relevant for oviparous species. For instance, most bird embryos pip internally, gaining access to the air-space within their egg some time before cracking the shell; this allows them to begin breathing air and increase their oxygen uptake to support the exertion of external pipping and exit from the shell (*Oppenheim, 1972*; *Pettit & Whittow, 1982*; *Visschedijk, 1968a*;

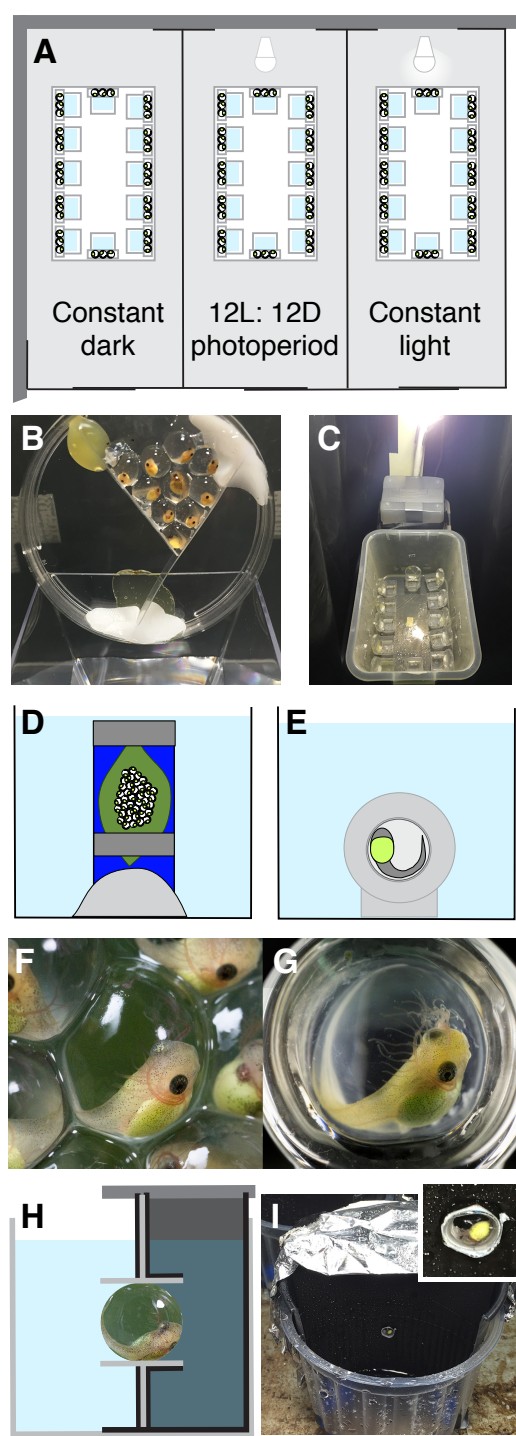

**Figure 1** **Experimental methods.** For the hatching timing experiment, we made three light-proof compartments (A), each with a different photoperiod. We split each *Agalychnis callidryas* egg clutch into three groups of embryos, held in a petri dish over dechlorinated tap water (B). We placed dishes of embryos within plastic bins in each compartment (C). 

**Figure 1 (…continued)**
To observe hatching orientation underwater, we submerged whole egg clutches (D, F) and individual embryos in glass cups (E, G). The surface exposure of eggs was similar but embryos' visual environments differed (F, G). In the hatching phototaxis experiment, we placed *A. callidryas* eggs in close-fitting tubes in the dividing wall of half-dark cups (H, I). Schematics are not to scale; compartments were longer than in A, aquaria larger than in D, E, and half-dark cups larger than in H (see I).

*Visschedijk, 1968b*; *Visschedijk, 1968c*). Some fish eggs are asymmetrically shaped and attached to substrates; at the time of hatching, embryos orient their heads to the free end of their egg to exit from it (*Korwin-Kossakowski, 2012*; *Olivotto et al., 2003*). Correct orientation appears essential for hatching to occur in several of these species. In amphibians, egg capsules lack opercula or rupture planes. In some species the entire vitelline membrane is enzymatically degraded (*Altig & McDiarmid, 1999*), while in others embryos digest only a small escape hole (*Cohen, Seid & Warkentin, 2016*). In both cases, embryos seem able to exit in any direction. Nonetheless, for eggs closely packed into gelatinous masses, hatching outward to directly enter the outside medium may be more efficient or successful than hatching toward the center of the mass and then moving through the matrix of jelly and other eggs to escape.

The typical structure of hatching-competent *A. callidryas* clutches is a monolayer of eggs adhered to a core of jelly adhered to a leaf, with each egg having part of its surface exposed to air and part pressed against other eggs and the underlying jelly (*Gomez-Mestre, Wiens & Warkentin, 2008*; *Warkentin, 2000a*; *Warkentin, 2002*). Thus, eggs have strong internal oxygen gradients with the highest concentration of oxygen at the air-exposed surface (*Warkentin, Gomez-Mestre & McDaniel, 2005*). As early as the neural tube stage, embryos orient their developing heads towards the air and reposition themselves after being displaced, and more developed embryos spend most of their time with their external gills toward the air (*Rogge & Warkentin, 2008*). The fact that embryos orient in their egg's oxygen gradient means they are usually positioned to hatch outward, into the air, without any active orientation at hatching. However, one common threat to eggs, flooding, disrupts this orientation. For *A. callidryas*, as for many species with terrestrial egg masses, flooding causes hypoxia, which stimulates embryos to hatch (*Doody et al., 2001*; *Martin, 1999*; *Petranka, Just & Crawford, 1982*; *Warkentin, 2002*). However, the initial response of *A. callidryas* embryos to flooding is to change position, many times, thus disrupting their outward-facing orientation (*Rogge & Warkentin, 2008*). Therefore, to hatch into the water, instead of into jelly or other eggs, flooded *A. callidryas* embryos presumably need another orientation cue. Thus, a second context in which visual or light cues may be important for *A. callidryas* embryos is to re-establish an appropriate hatching orientation when flooded.

Our study of hatching orientation was motivated by observations that hatching was frequently mis-oriented for embryos flooded individually in glass cups (for another study). The air-exposed surface of eggs in cups is similar to that in clutches, maintaining a natural oxygen environment, but in other ways, including visually, their environment differs substantially from that in natural clutches (Figs. 1F–1G). Thus, to assess if something about the natural clutch environment provides a directional cue that allows embryos to re-establish their outward orientation for hatching, after it is disrupted, we flooded entire

clutches on leaves and recorded the hatching orientations of embryos (toward open water or not). Based on the difference in hatching orientation in cup vs. clutch contexts, and the generally darker, more light-blocking structure of leaves and eggs compared to glass, we hypothesized that in flooded clutches *A. callidryas* embryos use light cues, or positive phototaxis, to orient their hatching. To test this, we positioned individual eggs in tubes with one end illuminated and the other dark, then flooded them and recorded the direction of hatching.

## METHODS

### Egg collection and maintenance

Red-eyed treefrogs lay eggs at night, with most eggs laid between 22:00 and 02:00 h; thus we report clutch and embryo ages starting from midnight of their oviposition night. We collected 0–3 day old *A. callidryas* egg masses on leaves from the Experimental Pond (9°7′14.77″N, 79°42′12.03″W) in Gamboa, Panama, and brought them to an open-air, ambient temperature and humidity laboratory at the Smithsonian Tropical Research Institute. We attached leaves with clutches to plastic cards, for support, and removed any dead or abnormally developed embryos to ensure all embryos used in experiments were healthy and normally developed. We kept clutches in cups over aged, dechlorinated tap water in large, mesh-lidded bins and misted them frequently with rainwater to maintain high humidity and clutch hydration. After experiments, we returned all hatched tadpoles to the Experimental Pond. Experiments were conducted under permit SC/A-15-14 and SE/A-46-15 from the Panamanian Ministerio de Ambiente, STRI IACUC protocol 2014-0601-2017, and BU IACUC protocol 14-008.

### Hatching timing experiment

We used a split-clutch design to assess the role of light cues in diel hatching rhythms of *A. callidryas* embryos from August 4 to 7, 2015 (Figs. 1A–1C). We divided each clutch into three groups of 7–10 embryos, exposed them to three different light environments (continuous light, continuous darkness, and a 12L:12D photoperiod; 108 eggs per treatment), and recorded when embryos hatched. We initially set up 12 clutches, but a subset of one clutch in the continuous dark treatment dried, and hatched early, so we restrict our analyses to the 11 clutches whose hatching patterns were unaffected by any direct risk factor ($N = 11$). Including the clutch with the dried subset in our analyses does not change the statistical significance, or non-significance, of any result.

We divided clutches and set up the treatments at 18:00 h when embryos were 3 days old, before physical disturbance induces hatching (*Warkentin et al., 2017*). To support each group of eggs, we glued two strips of plastic (30–45 mm long) into a 47 mm diameter plastic petri dish, creating a V-shaped compartment, attached the dish at a 75-degree angle to the inner wall of a square plastic cup, and lined the egg compartment with enough jelly from the clutch to provide a hydrated foundation for the designated number of eggs (Fig. 1B). We gently removed each egg from the clutch using curved, blunt-ended forceps and positioned it on jelly in the petri dish. We arranged eggs in a monolayer, keeping surface exposure to air within the natural range (*Warkentin, 2000b*; *Warkentin,*

*2002*). We used black plastic sheeting to construct three lightproof compartments, one for each treatment, within a small room in the ambient laboratory (Fig. 1A). For the dark treatment and during the dark period of the photoperiod treatment compartments were completely dark, except during observations which we made under dim red light. We placed one group of eggs per clutch into each compartment, inside a $36 \times 22 \times 23$ cm plastic bin with a mesh lid to maintain humidity (Fig. 1C). We wrapped bins in black plastic and, for extra security in the continuous dark and photoperiod treatments, we constructed screened portions of lids using multiple layers of black mesh to allow gas exchange but reduce light entry. For the continuous light treatment and the light period of the photoperiod treatment, we used lids with a single layer of light green mesh that allowed light penetration. We checked eggs every 2 h from 06:00 h, 4 d, until all embryos had hatched. At each time point, we carefully removed the lids, which we designed for easy removal and replacement with minimal vibrations. We counted the unhatched eggs in each dish and/or the hatched tadpoles in each cup, and carefully misted the walls and lid of each container to maintain humidity without disturbing the embryos. Then we replaced the lids, changing the lid in the photoperiod treatment every 12 h when lights switched on or off. For the continuous light treatment and the light period of the photoperiod treatment (06:00–18:00 h), we illuminated compartments with a single LED light (700 lumens, 6500 K, LDALV9D67ML, Cool Daylight LED, Panasonic) suspended at the same height (ca. 50 cm) over each bin (Fig. 1C). We spot-checked temperatures within compartments 3 times during the experiment, recording a range from 27.4 to 28.0 °C.

A limitation of our experiment, imposed by field-season logistics, is that we only had one light-controlled compartment per treatment; thus, any other environmental variation between compartments was confounded with light treatment. In general, variation in temperature and humidity can influence the hatching timing of amphibians via effects on development rate (*Duellman & Trueb, 1986*; *Seymour & Bradford, 1995*). In *A. callidryas* these variables may also directly affect embryos' hatching decisions; both drying and heating can induce hatching (*Salica, Vonesh & Warkentin, 2017*; SC Guevara-Molina & K Warkentin, 2018, unpublished data). For instance, during the warm El Niño conditions in 2015, embryos in our laboratory developed faster and hatched earlier than in other years (see *Warkentin et al., 2017*). All treatments were run simultaneously, on split-clutches, and eggs in different compartments were separated by only a short distance and thin plastic sheet. Thus they were subjected concurrently to any overall temperature fluctuations in the room. We used LED lights to minimize heat from the bulbs and manual misting to maintain humidity. In our few spot checks, all compartments were within a range of 0.6 °C, but we do not have continuous temperature data through the experiment.

## Hatching orientation
### Does the natural clutch environment provide directional cues after flooding?

We initially noticed a high frequency of mis-oriented hatching by embryos flooded outside their natural clutch environment. In 2014, as part of a study on the ontogeny of hatching performance in *A. callidryas* (K Warkentin et al., 2014, unpublished data) we

placed individual embryos in glass cups, flooded them, and recorded their response using macro-videography ($N = 21$, 4-day old embryos). At this developmental stage embryos hatch readily in response to multiple cue types, including hypoxia, and have large, well developed eyes (*Warkentin, 1995*; *Warkentin, 2002*; *Warkentin et al., 2017*). Glass cups (interior diameter 5 mm, depth 3–4 mm, Fiamma Glass, Waltham, MA; Figs. 1E and 1G) were custom made to fit eggs closely and provide natural surface-area exposure (*Rogge & Warkentin, 2008*; *Warkentin, 2002*). We placed eggs in cups at age 3 days, then placed eggs-in-cups in humidors and misted them often with rain water to maintain hydration. For each test, we moved a single egg-in-cup into a small aquarium, waited five minutes for acclimation, then gently and slowly flooded the aquarium to submerge the embryo. To minimize the time for boundary layer formation and local oxygen depletion around eggs, we used hypoxic water. Water was degassed by boiling it for at least 10 min, allowed to cool in sealed glass jars, and used within 30 min of opening each jar ($15 \pm 1.3\%$ air saturated at opening to $21 \pm 3.0\%$ air saturated at 30 min, $N = 10$ jars). Embryos were video-recorded using a Canon EOS 5D Mark III camera and MPE-65 mm macro lens until they hatched. Hatching direction was recorded from the videos.

Embryos flooded individually in glass cups had a natural oxygen environment, but an unusual visual environment, since light could enter from all sides (Figs. 1F vs. 1G). To quantify the frequency of mis-oriented hatching for embryos flooded in their natural clutch-on-leaf context, and assess if that environment may provide some directional cue for hatching embryos, we recorded hatching orientation (into open water or not) under whole-clutch flooding. We conducted whole-clutch flooding experiments from July 12 to 24, 2015, using 4 day old clutches ($N = 14$). Clutches were tested individually or in pairs. For each clutch, we used Plasticene to hold its plastic support card vertically at the bottom of a small aquarium (Fig. 1D). Cards were positioned parallel to the aquarium wall, ca. 5–8 cm away, so that embryos were easily visible and would emerge fully from the clutch before contacting the wall. We waited five minutes after setting up each clutch, and no embryos hatched in response to the set-up process. We then flooded the aquarium with aged, de-chlorinated tap water (initially fully air saturated) to submerge the entire clutch. We poured water slowly and gently to minimize vibration, using a funnel attached to a long pipette stem to introduce water into the bottom of the tank. We then continuously observed the embryos, recording hatching direction until all embryos had hatched, or for 60 min if not all hatched. A few embryos failed to exit their egg capsule through their initial membrane-rupture site, remaining within collapsed vitelline membranes for periods of time. For these individuals, we could not accurately assess the directionality of their initial hatching attempt; thus, we restricted our analyses to embryos that exited through their first hole. We compared the incidence of hatching into open water vs. hatching into the clutch (i.e., into the jelly behind eggs).

### Do submerged embryos use light as a cue for hatching orientation?

The poorer hatching orientation we observed in glass cups suggests that any potential memory of embryo's prior oxygen gradient is insufficient as a cue, and that visual cues may be relevant. Thus, to test *A. callidryas* embryos for a contribution of light or visual

cues to hatching orientation, we conducted hatching phototaxis experiments at 4 days of age from 12:00 to 17:00 h. We held each test egg in a small tube set into the dividing wall of a half-dark cup, so that one direction was illuminated and the other dark, then flooded it with hypoxic water and recorded hatching direction (Figs. 1H–1I). We constructed 20 half-dark cups and tested embryos in batches. To make each cup, we inserted half of a black plastic cup into a clear plastic cup and positioned a vertical plastic wall centrally to separate the black cup inset and clear cup sections. We encased the dividing wall and sealed it to the cup floor and wall with lightproof black duct tape. We also covered half of the outside of the clear cup with black duct tape, on the dark side, and covered the top of the dark side with aluminum foil to prevent light from entering. In the center of the dividing wall of each cup, we made a small hole and secured a 5 mm long piece of clear plastic tube, cut from a 5 mm inner diameter pipette, orienting it horizontally. At age 3 days, prior to the onset of hatching responses to physical disturbance (Warkentin et al., 2017), we gently removed individual eggs from their clutches and placed each one into a tube in a half-dark cup, alternating side of egg insertion (dark or light), then covered the dark side. We put all the half-dark cups with age-matched embryos together in a plastic bin, left embryos undisturbed overnight, and tested them the next day.

We tested 92 embryos for hatching orientation, from 39 clutches (up to 3 siblings per clutch), running 10–15 trials concurrently in eight sets from July 4 to 20, 2015. Seven trials were eliminated from analysis because eggs slipped out of their tubes when flooded, leaving $N = 85$. As a control for side-bias due to some element of the physical structure of the cups, unrelated to light or visual cues, we tested 20 additional embryos, from 5 clutches, in a dark room after a twenty minute dark-acclimation period, using only dim red light to observe hatching direction. The way that we constructed the cups precluded an equivalent "both sides light" control to test for side-bias due to aspects of the visual environment. Thus, our experiment does not address whether embryos might tend to orient hatching toward or away from any particular visual stimulus, simply whether they orient toward light and visual cues rather than their absence.

Because eggs experience a wide range of light environments in the wild, from direct sunlight to deep shade, we tested hatching orientation under two light intensities (higher and lower, trials 1–50 and 51–92, respectively); to reduce illumination we increased the distance to the light source and wrapped black tape around the egg tube. All embryos were exposed to natural light from windows and overhead florescent lab lighting. Before starting each experimental trial set, we increased this illumination using two LED lights (320 lumens, 5000 K, 2CYSVB; Ottlight Technologies, Tampa, FL, USA) positioned at a distance of about 10 cm (higher illumination, trials 1–50) or 15–20 cm (lower illumination, trials 51–92) above the cups. We did not measure light intensity in this experiment, nor do we have field measurements of underwater light intensity in *A. callidryas* ponds. After switching on the LED lights, we gave embryos three minutes to acclimate to the light environment, then gently flooded their cups with hypoxic water. We observed all trials continuously until embryos had hatched, noting hatching direction and timing. Because eggs fit closely inside tubes, embryos were constrained to hatch either toward the light

or dark, and exited their egg and tube simultaneously. Thereafter, hatched tadpoles were confined to one half of the cup, allowing us to confirm hatching direction after trials.

## Statistical analysis

We performed all our statistical analysis using R v3.3.1 (*R Core Development Team, 2017*). We used the package "lme4" (*Bates et al., 2015*) to analyze hatching over time and by treatment. We used a generalized linear mixed-effects model (GLMM) with an underlying binomial error structure and logit link function to test for differences in hatching timing of embryos across different light treatments. The binomial response variable was the number of tadpoles hatched at each time point from a petri dish (i.e., a clutch subset within a treatment), out of the initial number present in the dish; for visualization we graph this as a proportion. We included age, treatment, and their interaction as fixed predictors, and clutch as a random effect to account for non-independence of observations of embryos in the same clutch. We then compared increasingly parsimonious, nested models with likelihood ratio tests to estimate *p* values of the predictors and their interactions. Tukey's post-hoc pairwise comparisons among light treatments were made using the glht (general linear hypothesis test) function in the "multcomp" package (*Hothorn, Bretz & Westfall, 2008*). We used ANOVAs with Tukey's post-hoc pairwise comparisons to test for a treatment effect on the proportion of embryos hatched just after the onset of darkness at age 5.75 d and on the level of hatching synchrony *within* egg masses (dishes) estimated as the maximum proportion of embryos that hatched during any 2-h period. We used Shapiro–Wilk normality tests on the residuals of each fitted model to ensure parametric tests were appropriate; non-parametric tests were used when the data did not meet the requirements for parametric statistics. We used Levene's tests for homogeneity of variance to determine the effects of light treatment on hatching synchrony *among* egg masses (dishes), i.e., how much did the timing of peak hatching vary among sub-clutches within each treatment. Because some dishes had equally high numbers of embryos hatch during two different 2-h intervals, we used 4-h intervals to determine a single peak (modal) time of hatching for each mass. We then compared the distributions of these modal hatching times across treatments. We used Kruskal–Wallis tests followed by Dunn's tests of multiple comparisons to compare the proportion of embryos hatched by age 5.75 d and after age 6.0 d in each treatment to assess whether continuous darkness accelerated hatching and continuous light delayed hatching.

We used a test of equal proportions (prop.test function) to test for a difference in the incidence of hatching into the glass or jelly between individual egg and whole-clutch flooding experiments, directly comparing the observed proportions of mis-oriented hatching in these two contexts. We then tested for non-random hatching orientation in flooded whole clutches using a test of equal proportions, based on the null hypothesis that the probability of hatching into open water by chance is equal to the proportion of exposed surface area of embryos. Because we did not have individual measures of surface exposure, we used both extremes of the natural range (25–50%; *Warkentin, 2000a*; *Warkentin, 2002*).

For embryos in half-dark cups, we tested for non-random hatching orientation using a test of equal proportions, based on the null hypothesis that the probability of hatching in each direction was 50%. We used a GLMM to test whether the 'side of insertion into the

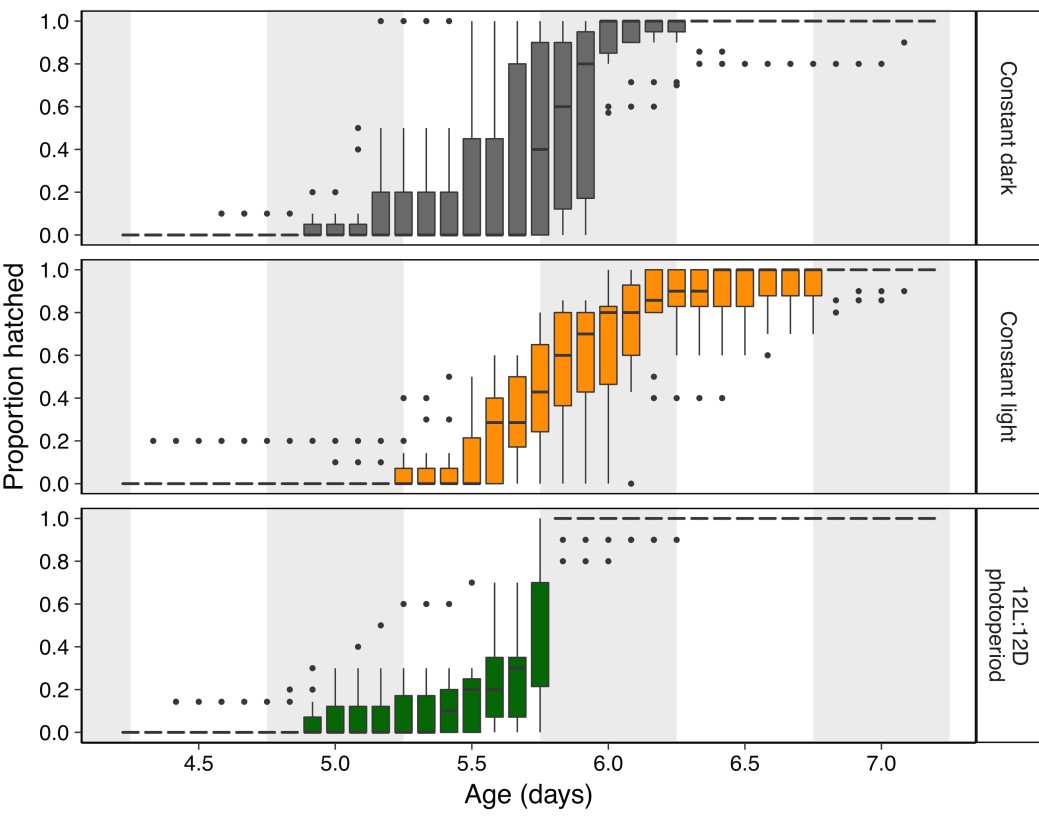

**Figure 2  Hatching timing of *Agalychnis callidryas* embryos in three light environments.** Data are proportion hatched for embryos in each treatment, recorded every 2 h ($N = 11$ clutches, each split across treatments). Boxplots show median, interquartile range (IQR), and extent of data to $\pm 1.5 \times$ IQR (box and whiskers), and outliers. Grey background sections indicate dark periods in the photoperiod treatment, and the corresponding periods of time in constant dark and constant light treatments.

tube' affected hatching direction, including clutch as a random factor ($N = 57$ trials for which side of insertion was recorded). For all statistical tests, $\alpha = 0.05$.

## RESULTS

### Effect of light-dark cycle on diel hatching pattern

Light environments affected hatching timing of *A. callidryas* embryos. Embryos hatched more gradually and later in continuous light, showed more variation in hatching timing among clutches in continuous darkness, and showed the strongest hatching synchrony (i.e., hatching concentrated within a short period of time) in the 12L:12D photoperiod, with a sharp peak of hatching shortly after dark at 5.75 days. All tested embryos hatched successfully. Overall, hatching in all treatments was gradual and began at 4 days of age with spontaneous hatching of few individuals (Fig. 2). Clutches reached 50% hatched at 5 days in all treatments. They completed hatching earlier in the photoperiod treatment, at 6 days, and later in both other treatments, at 7 days (Fig. 2). Analyzing the entire hatching curves using a GLMM, we found that hatching timing was influenced by age,

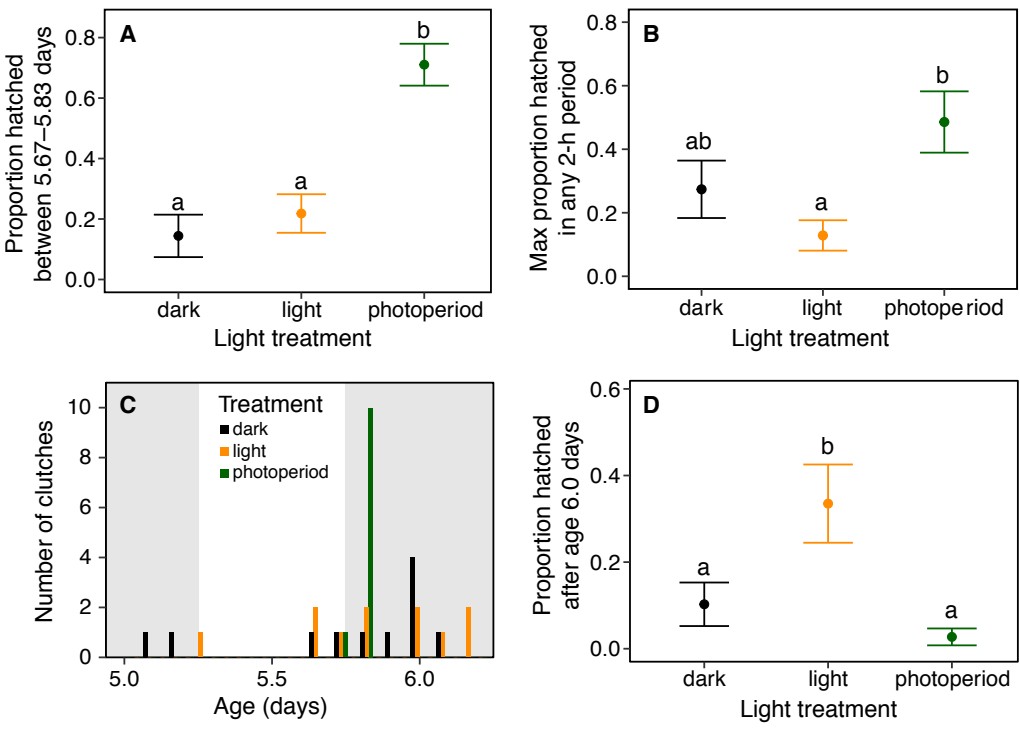

**Figure 3** Timing and synchrony of hatching of *Agalychnis callidryas* embryos in three light environments, for N = 11 clutches, each split across treatments. (A) Hatching at nightfall. Data are mean proportion of embryos that hatched between 16:00 and 20:00 h at age 5 d, ±SE across sibships. Hatching was more concentrated in this 4-h period when it included the onset of darkness; different letters indicate significant differences at α = 0.05. (B) Hatching synchrony within egg masses, measured as the highest proportion of embryos hatched in any 2-h period. Embryos within masses hatched more synchronously in the photoperiod treatment. (C) Synchrony across egg masses (sibships). Data are number of clutches whose peak of hatching occurred in each 4-h time period. Grey bars indicate dark periods of photoperiod treatment. Hatching was more synchronous across sibships in the photoperiod treatment. (D) Delayed hatching. Proportion of embryos that hatched after age 6 d. More embryos hatched late in development in the continuous light treatment.

treatment, and a time-by-treatment interaction (Fig. 2; GLMM, age $\chi^2 = 10,317$, $p < 0.001$; treatment $\chi^2 = 272.56$, $p < 0.001$; interaction $\chi^2 = 92.442$, $p < 0.001$). Unsurprisingly, the proportion hatched increased over time in all treatments; all embryos hatched by 04:00 h, 7 d. Post hoc tests revealed differences in the proportion of embryos hatched over time between the photoperiod and both light and dark treatments (Tukey's post hoc tests, photoperiod vs. light; $p < 0.001$; photoperiod vs dark $p < 0.001$). Embryos in the photoperiod treatment had pulsed, synchronous hatching shortly after the onset of darkness at age 5.75 d; ca. 71% of all embryos in the photoperiod treatment hatched immediately after the onset of darkness (vs. 21% and 14% during the same time period in the light and dark treatments, respectively; Fig. 3A; ANOVA, $F_{2,30} = 20.596$, $p < 0.001$; Tukey's post hoc tests, photoperiod vs. light $p < 0.001$; photoperiod vs. dark $p < 0.001$). Hatching synchrony *within* egg masses was highest in the photoperiod treatment (Fig. 3B; highest proportion hatched during any two-hour period, ANOVA, $F_{2,30} = 5.6252$, $p = 0.008$; Tukey's post

hoc tests, photoperiod vs. light $p = 0.006$; photoperiod vs. dark $p = 0.135$). Moreover, the photoperiod treatment also had the most synchronous hatching *across* egg masses. Clutches in the photoperiod treatment clearly had more similar modal hatching times compared to clutches in other treatments (Fig. 3C; Levene's test, photoperiod vs. light $F_{1,20} = 14.76$, $p = 0.001$ ; photoperiod vs. dark $F_{1,20} = 7.0203$, $p = 0.0154$ ; light vs. dark $F_{1,20} = 0.093$, $p = 0.7635$). We found no evidence that continuous darkness accelerated hatching; at age 5.75 days all treatments had a similar proportion of embryos hatched (44% dark, 45% light, 44% photoperiod; Kruskal–Wallis, $\chi^2 = 0.096$, $p = 0.953$). However, a larger proportion of embryos in the light treatment hatched after age 6.0 days compared to those in the other two treatments (Fig. 3D; Kruskal–Wallis, $\chi^2 = 11.476$, $p = 0.003$; Dunn's post hoc tests, light vs. dark $p = 0.027$, light vs. photoperiod $p = 0.003$).

### Hatching orientation

When flooded in glass cups, 28.6% of embryos attempted to hatch toward the glass, rather than through the exposed portion of their egg surface. In contrast, when embryos were flooded in whole clutches, we found that only $3.2 \pm 4.2\%$ (mean $\pm$ SD across clutches) of embryos emerged into the jelly rather than toward the open water. Using a test of equal proportions, we found a significant difference in the proportion of poorly oriented hatching between these two contexts ($\chi^2 = 35.834$, $df = 1$, $p < 0.001$). The proportion of embryos in whole-clutch flooding experiments that hatched toward open water ($96.6 \pm 4.5\%$, mean $\pm$ SD across clutches), was significantly higher than that expected by chance based on the natural range of exposed surface area for individual eggs (25%: $\chi^2 = 6783.2$, $df = 1$, $p < 0.001$; 50%: $\chi^2 = 3297.3$, $df = 1$, $p < 0.001$).

We found no evidence of a light-side hatching bias due to the physical structure of our half-dark cups, in the absence of illumination by white light. In a dark room, 9 embryos hatched towards the open "light" side and 11 towards the covered "dark" side of cups ($\chi^2 = 0.05$, $df = 1$, $p = 0.823$). However, when the cups were illuminated, *A. callidryas* embryos were more likely to hatch towards the light. In both sets of experimental trials, with higher and lower illumination, significantly more embryos hatched toward the light (trials 1–50: 67%, $\chi^2 = 4.356$, $df = 1$, $p = 0.037$; trials 51–92, 73%, $\chi^2 = 7.225$, $df = 1$, $p = 0.007$). We found no difference between the two sets of illuminated trials ($\chi^2 = 0.611$, $df = 1$, $p = 0.434$); overall, 59 of 85 tested embryos hatched toward the light (69.4%, Fig. 4; $\chi^2 = 12.047$, $df = 1$, $p < 0.001$). Embryos hatched $16.27 \pm 13.24$ min after flooding (mean $\pm$ SD; range 3.07–63.28 min) and, as expected, the initial side of insertion into the tube had no effect on hatching direction (GLMM, $\chi^2 = 1.799$, $p = 0.407$).

## DISCUSSION

Our results support that embryos of *A. callidryas* use light cues to inform their hatching in two different ways. First, under natural photoperiods, embryos showed a clear diel pattern of hatching, with a strong peak shortly after nightfall. Their hatching was desynchronized by continuous light and darkness and delayed by continuous light but not accelerated by continuous darkness. Second, embryos appear to use light cues to orient hatching from flooded eggs, showing effective positive phototaxis when oxygen gradient cues

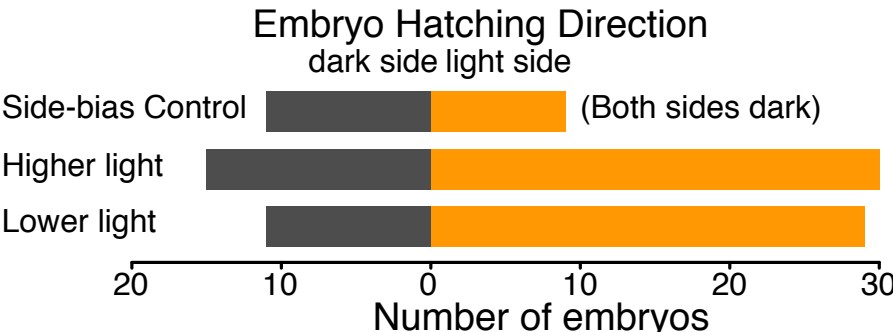

**Figure 4** Hatching direction of *Agalychnis callidryas* embryos submerged in hypoxic water with one side illuminated (phototaxis experiments) or neither side illuminated (control). In the absence of light, there was no evidence of side-bias due to the structure of the half-dark cups. Under two illumination intensities more embryos hatched towards the light. Data are number of embryos exiting their egg, and egg-holding tube, in each direction.

are disrupted. This adds another sensory modality to the information sources *A. callidryas* embryos use to inform their hatching process, and another aspect of hatching—orientation as well as timing—that is responsive to environmental cues.

## Diel pattern of hatching

Could the differences in hatching pattern we observed between our light-cycle treatments be caused by something other than the designed differences in illumination? The hatching patterns we observed are directly opposed to expectations based on unintended heating or drying effects of the lighting. First, the embryos in continuous light showed delayed hatching, but from either heating or drying effects of the lighting we would expect accelerated hatching. Second, we only observed signs of dehydration in one sub-clutch (dish), and that was in the continuous dark treatment, subjected to the lowest potential for unintended heating/drying effects of lights. Third, any heating or drying effects of the lighting within the photoperiod treatment would be expected to increase hatching during the light period, not during the dark period. For these reasons, we consider the observed differences in hatching pattern to reflect effects of the intended differences in light environments, not unintended differences in temperature or humidity. Importantly, our photoperiod treatment recovered the nocturnal peak of hatching that is well-documented in the field (*Gomez-Mestre & Warkentin, 2007*; *Warkentin, 1995*).

Embryos with plastic hatching timing can make crucial and often swift decisions to leave their egg to escape immediate threats (*Warkentin, 1995*; *Warkentin, 2000b*). Thus, diel hatching patterns are likely most relevant for undisturbed egg clutches, in which embryos apparently hatch spontaneously. In *A. callidryas*, most undisturbed embryos hatch at night (*Gomez-Mestre & Warkentin, 2007*; *Warkentin, 1995*). Nocturnal hatching may be favorable for embryos of many species, since hatchlings are often susceptible to diurnal visual predators (*Bradbury et al., 2004*; *Gustafson-Marjanen & Dowse, 1983*; *Touchon et al., 2013*; *Warkentin, 1995*; *Witherington, Bjorndal & McCabe, 1990*). This is consistent with an inhibitory or delaying effect of light on hatching, as is evident in *A. callidryas* and some fish

species (e.g., *Brüning, Hölker & Wolter, 2011*; *Downing & Litvak, 2002*). In addition, diel hatching patterns may be particularly valuable for species which exhibit habitat switches at the time of hatching, with different diel patterns of risk in each habitat. For instance, a generally higher risk from terrestrial egg-predators at night, and from aquatic predators of tadpoles during the day, could favor hatching at dusk (e.g., nocturnal snakes vs. diurnal fishes; *Gomez-Mestre, Wiens & Warkentin, 2008*).

Under natural photoperiods, *A. callidryas* embryos showed a clear peak of hatching (i.e., relative synchrony) shortly after the onset of darkness. Synchronous hatching may reflect a simple convergence of many embryos on a shared optimal hatching time. Synchrony could also, in itself, be beneficial. Synchronous hatching and synchronous emergence from nests have been suggested to reduce predation through predator-swamping or avoidance (*Christy, 2003*; *Ims, 1990*). If predators are alerted by initial hatchlings and converge on sites where hatchlings are emerging, higher predation on later-hatching individuals could also favor synchrony (*Ims, 1990*; *Testa, 2002*; *Tucker, Paukstis & Janzen, 2007*). Observations of fish predation on hatchlings suggest that hatching after dark may improve *A. callidryas* survival in some contexts (M Hughey & K Warkentin, 2007, unpublished data); however, to our knowledge relatively little research has examined diel patterns of tadpole predation.

Synchronous, nocturnal hatching is clearly a potential adaptive strategy for many species, and likely also relevant to *A. callidryas*. This strategy has been widely documented in fishes (e.g., tropical reef damselfish (*Asoh & Yoshikawa, 2002*; *Doherty, 1983*; *McAlary & McFarland, 1993*); Baltic salmon, *Salmo salar L.* (*Brännäs, 1987*; *Gustafson-Marjanen & Dowse, 1983*); rainbow smelt, *Osmerus mordax* (*Bradbury et al., 2004*)). Across different photoperiods, these species all show synchronous hatching shortly after the onset of darkness. This pattern is hypothesized to be a predator avoidance strategy that evolved due to high risk of predation when fry emerge. However, very few studies have directly, empirically linked this hatching pattern to diel variation in predation pressure (*Bradbury et al., 2004*). In the case of reptiles, most species develop in nests with temperature gradients which influence incubation and hatching times; thus, apart from some well documented cases in sea turtles (*Spencer & Janzen, 2011*) and pig-nosed turtles, *Carettochelys insculpta* (*Doody et al., 2001*), synchronous hatching is less common since hatching prematurely has immediate and long term costs on individuals (*Colbert, Spencer & Janzen, 2010*; *Doody, 2011*). A review of the timing of larval release in brachyuran crabs found that of the 81 species examined, 78 release larvae at or after sunset, and mostly during high amplitude high tides, presumably as a predator avoidance strategy (*Christy, 2003*; *Christy, 2011*). Amphibians provide some of the best-studied examples of environmentally cued hatching, with documented cases of hatching plasticity widely distributed across the clade (*Warkentin, 2011b*); yet, to our knowledge, our study is the first to test for a shift in hatching timing in response to light cues.

Our finding of reduced hatching synchrony in continuous darkness, but no general acceleration, suggests that the change from light to darkness, rather than darkness *per se*, may serve as a hatching cue in *A. callidryas*. Similarly, Baltic salmon (*Salmo salar L.*) embryos kept in dark conditions hatched continuously and unsynchronized; 50% of all eggs hatched within 6 days after the onset of hatching competence, while embryos in a

16L:8D photoperiod environment hatched more synchronously; 50% of eggs hatched within 2 days after the onset of hatching competence, with peaks of hatching after the onset of darkness (*Brännäs, 1987*). However, embryo responses to photoperiod may differ among species. For instance, hatching of haddock (*Melanogrammus aeglefinus*) embryos is delayed in darkness and advanced in light (*Downing & Litvak, 2002*). In freshwater fish, continuous light delayed hatching in roach (*Rutilus rutilus*) and bleak (*Alburnus alburnus*), but accelerated hatching timing in chub (*Leuciscus cephalus*), and only in one species, perch (*Perca fluviatili*), was the onset of darkness found to elicit more hatching (*Brüning, Hölker & Wolter, 2011*). Clearly, the specific effects of light, darkness, and light-dark transitions on hatching timing vary among species. Further examination of both the adaptive significance of diel hatching patterns and the underlying mechanisms eliciting these responses in light- or dark-cued species is needed.

Undisturbed hatching may appear spontaneous in red-eyed treefrogs, but it is clearly not simply a developmental process under full endogenous control. Rather, like induced early hatching, its timing is environmentally informed in ways that may confer selective benefits. This may also be true for apparently spontaneous hatching in other species as well.

## Hatching orientation

Our whole-clutch flooding experiment unequivocally demonstrates that *A. callidryas* embryos are able to orient hatching into the open, even when submerged. Might there be anything other than visual cues that could explain the hatching orientation we observed in our half-dark cups? The fact that we observed no "light-side" bias when embryos were tested in a dark room (i.e., both sides were dark) supports that non-visual aspects of the cup environment cannot explain the observed pattern; embryos did not avoid the "dark" side of the cup nor were they attracted to the "light" side in the absence of visual cues. Moreover, the similar response of embryos to two experimental light intensities suggests that the response is not limited to one specific, narrow range of lighting conditions.

For some embryos, the physical structure of their egg dictates hatching direction. For instance, in non-spherical eggs, egg shape may limit the range of positions embryos can occupy, and thus their potential hatching orientation (e.g., some demersal fish eggs, *Korwin-Kossakowski, 2012*; *Olivotto et al., 2003*). Other embryos, including *A. callidryas* and other amphibians, move freely within their vitelline chamber and could potentially exit through any part of their capsule. Their hatching orientation may, nonetheless, affect fitness. Specifically, for terrestrial eggs laid in masses on solid substrates, embryos that hatch toward the interior of their mass may become trapped between sibling eggs and the substrate. In our field and laboratory observations of *A. callidryas*, such cases are relatively rare but can be fatal (B Güell & K Warkentin, pers. obs., 1992–2018), revealing the importance of hatching orientation. Thus, there must be cues that allow embryos to hatch correctly oriented. Prior work has demonstrated that *A. callidryas* embryos in clutches in air orient toward the exposed surface of their egg capsules, presumably in response to the oxygen distribution within eggs (*Rogge & Warkentin, 2008*). Our results clearly demonstrate that these embryos are able to appropriately orient hatching when

submerged, and support that they use light cues in this context. This adds another sensory modality to the suite of cues that affect *A. callidryas* hatching behavior, beyond oxygen and mechanosensory cues (*Warkentin, 2002*; *Warkentin, 2005*). It also raises the possibility that oxygen gradients may not be the only cue informing *A. callidryas* orientation and hatching direction in clutches in air. Moreover, it suggests that other species might use light—or more complex visual cues—to inform hatching in some way beyond diel cycles.

Embryonic visual learning and its effect on post-hatching behavior has been demonstrated in several animal taxa (e.g., cuttlefish, *Darmaillacq, Lesimple & Dickel, 2008*; bobwhite quail, *Honeycutt & Lickliter, 2002*; leopard gecko, *Sleigh & Birchard, 2001*). These studies suggest that embryos may specifically receive and retain visual information presented to them *in ovo*, but not necessarily that they use this information prior to hatching. Human fetuses have also been shown to respond behaviorally to face-like visual patterns presented *in utero*, more than to inverted patterns (*Reid et al., 2017*). Red-eyed treefrog embryos clearly respond behaviorally to patterns of illumination around their eggs in at least one context, flooding. However, our study does not address to what extent these embryos are capable of perceiving images, such as embryos in neighboring eggs, the leaf to which the clutch is attached, or predators nearby. Nor does it address if, beyond simple light and oxygen gradients, they might use such visual information to inform hatching. However, these are now open questions.

Clearly the hatching response of *A. callidryas* embryos to light is context-dependent, reversing in polarity between the two cases we studied. In flooded clutches, *A. callidryas* embryos hatch toward light, or away from the dark, thereby avoiding potentially fatal hatching complications. In contrast, for the more common context of embryos in air, light inhibits or delays hatching and the onset of darkness appears to stimulate hatching, which might reduce larval predation risk. We know *A. callidryas* embryos show context-dependent cue use in at least two ecologically relevant contexts using different sensory modalities (*Warkentin, 2002*; *Warkentin, 2005*; *Warkentin, Caldwell & McDaniel, 2006*). Here we show that the presumably adaptive use of a novel cue type by *A. callidryas* embryos is also context-dependent.

## Conclusions

Like threat-induced early hatching, the timing of undisturbed, apparently spontaneous hatching in *A. callidryas* is environmentally cued. In undisturbed egg clutches and in flooded clutches suffering hypoxia, embryos use light cues in two different ways, to inform when and where to hatch. These findings add to the range of sensory modalities these embryos use to guide their behavior and support the generality of context-dependent cue use across sensory modalities. We propose that further investigation of how embryos use light and visual cues in hatching is worthwhile.

## ACKNOWLEDGEMENTS

We thank J Cuccaro for recording the 2014 videos, J Fumo for helpful input, O Wilson, P Gomez, A Spalding, J Shurin, and C Kurle for support, and members of the Egg Science Reading Group at Boston University for comments on the manuscript.

### Funding

This work was funded by the Smithsonian Tropical Research Institute (STRI), the National Science Foundation (IOS-1354072 to KMW and STRI-REU DBI-1359299 to BAG), and Boston University. The funders had no role in study design, data collection and analysis, decision to publish, or preparation of the manuscript.

### Grant Disclosures

The following grant information was disclosed by the authors:
Smithsonian Tropical Research Institute (STRI).
National Science Foundation: IOS-1354072 to KMW, STRI-REU DBI-1359299 to BAG.
Boston University.

### Competing Interests

The authors declare there are no competing interests.

### Author Contributions

- Brandon A. Güell conceived and designed the experiments, performed the experiments, analyzed the data, prepared figures and/or tables, authored or reviewed drafts of the paper, approved the final draft.
- Karen M. Warkentin conceived and designed the experiments, analyzed the data, contributed reagents/materials/analysis tools, prepared figures and/or tables, authored or reviewed drafts of the paper.

### Animal Ethics

The following information was supplied relating to ethical approvals (i.e., approving body and any reference numbers):

Smithsonian Tropical Research Institute and Boston Univeristy (STRI IACUC protocol 2014-0601-2017 and BU IACUC protocol 14-008).

### Field Study Permissions

The following information was supplied relating to field study approvals (i.e., approving body and any reference numbers):

Experiments were conducted under permit SC/A-15-14 and SE/A-46-15 from the Panamanian Ministerio de Ambiente.

### Data Availability

GitHub: https://github.com/bguell/Light-Cue-Manuscript-Final/tree/master.

### Supplemental Information

Supplemental information for this article can be found online at http://dx.doi.org/10.7717/peerj.6018#supplemental-information.

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
