# Peer review of "When and where to hatch? Red-eyed treefrog embryos use light cues in two contexts"

_PeerJ, doi:10.7717/peerj.6018_

## Round 0.1 · original submission · Major Revisions

We have now received the referees' comments on your manuscript . As you will note, the reviewers expressed several problems that must be taken into consideration. Both agreed in that your experimental design is not clear enough and need to be improved accordingly. Concerns have also raised about your statistical analyses and the organization and quality of your figures. I am sure that if you take all their suggestions in full consideration the quality of your manuscript will be highly improved. Please revise the manuscript in accordance with their advice.

Reviewer 1 ·

Basic reporting

'no comment'

Experimental design

Experiments description, in particular Hatching orientation section is a little confuse (lines 227-251) and could be improved for clarity. I suggest to divide this section (Hatching orientation) by following the two big questions tested i- To test whether environment provides cue for hatching timing and ii- to test for light-cued hatching orientation. I think authors should state at the beginning of the second question (ii-), as shortly and clearly as possible a description of Experiment 1 and Experiment 2. Perhaps, a figure showing the setup from dark and light source will help the reader to understand better the experimental design and interpret the results.

Validity of the findings

Regarding figures, I suggest to improve or show in a different manner results from Figure 3. It is a little confuse the inclusion of SE as shaded ribbons with different colors. And also will be fine to combine both, Fig 3 and Fig 4 in only one figure as is showed in Fig 5.

Also, I have some minor commentaries about figures:
Figures 1 and 2 should be better resume in one.
I suggest to delete Figure 7. Results are well described and sustained in text and by statistics.

Additional comments

The manuscript is well presented and structured. Language is clear and unambiguous permitting the reader to fluently follow the text.
Beyond there is many information about how different cues shift hatching in amphibians, in this paper authors present novel data on how an amphibian species Agalychnis callidryas use external cues to inform when and where to hatch based on light cues.
Introduction and background resulted clear, and then situating the reader in perspective through relevant and pertinent literature about hatching timing as response of direct threats. Also signaling the scarce information available on the spontaneous hatching or absence of immediate threats. Again, the manuscript result very valuable at this point.
I consider that the structure of the manuscript conforms the standards required by PeerJ following the forms usual for these kind of discipline.
Data analysis is well explained and statistically sound and I thank authors for providing raw data; these supplemental files helped to check in more detail results presented and also better interpret the Figures.
The research questions are well defined followed by an appropriate experimental design to respond the questions.
Finally, this work has novel results relative to hatching responses to cues not yet explored in amphibians The results are very clear and discussion is well supported by literature. Conclusions are well presented, following the initial questions and focusing on the supporting results. Also, the investigation was performed following ethical standards and permits (experiments conducted under specific permits detailed in the manuscript).

Reviewer 2 ·

Basic reporting

The language is ok but the manuscript suffers from a lack of clarity at several places (detailed in the “experimental design section”).
I would recommend the authors to present more deeply the findings coming from the same lab on hatching behaviour of red-eyed treefrogs. Particularly, I would recommend giving a more exhaustive review of environmental factors influencing hatching time and synchronicity in the introduction section (e.g. for example, the reference Salica et al., 2017 is missing). The background is very general and important informations on the biological model are missing.
I would recommend the authors to re-organise the introduction section in order to improve fluidity and the articulation between hypotheses. To that end, I would recommend removing the subheading “study organism and hypothesis” and, information given lines 97-113 should be placed after lines 72-73.
Lines 111-113: the hypotheses regarding hatching time and synchronicity should be clarified. Why the authors expected earlier hatching in the context of continuous dark? Does it means that the lack of light cues may accelerate embryonic development or engender hatching at an earlier stage of development? In the same direction, why did the authors expect an influence on the embryonic development of embryos in the context of continuous light? If an endogenous rhythm is expected by the authors (line 110), why did they not expect similar patterns of hatching between the contexts? This is confusing.
Lines 115-116: the authors should state more clearly that they investigated the influence of light in the specific context of environmental-induced (hypoxia) hatching and not just “when exiting the clutch”. The hypotheses tested should be clarified. The authors state line 92; “we examine hatching orientation and test the role of light as a possible orientation cue” and line 194: “To test if the natural clutch environment provides a cue….”. I did not get the relevance of using both embryos in clutches and in individual tubes. In my opinion there are too many uncontrolled environmental factors between the two conditions to make reliable comparisons. What was the hypothesis tested? Did the authors expect a higher success in whole clutches than in single embryos and why? If the authors just wanted to test the effect of light, then the focus should not be between “treatments” but within each treatment.
The images should be revised and their number reduced. I would recommend using a boxplot representation and not curves for the data presented Fig 3. The photographs presented in Figures 1 and 2 are nice but not informative enough. Schematic representations or pictures of the whole dispositive (and not just embryos) should be presented. As it is, it’s very difficult to understand how the embryos were maintained, particularly for the part related to hatching orientation. Fig 7 should be revised. The data should be presented as boxplots and the division of data in two sets (experiments 1 and 2) is confusing.
Raw data are supplied except for whole-clutch experiments.

Experimental design

The research is original and within the scope of the journal.
As explained above, the research questions are not well defined.
The methods are partly questionable and the paper generally suffers from a lack of clarity.
Possible methodological flaws need to be addressed.
Line 162: how were the embryos manipulated?
Lines 170-171: what is the link between the present study and the study mentioned (Warkentin et al., 2017)? Were the embryos exposed to other manipulations than those presented in the present paper?
Lines 173-174: was this protocol previously used?
Lines 176-190: the method is unclear. Was each treatment attributed to a specific room? Were the 12 clutches per treatment prepared and observed simultaneously? If different trials were conducted, this has to be mentioned and included in the analysis.
There is no indication of the temperature and the humidity within each room. This information is crucial since the results could be explained by a simple room-effect. As temperature and humidity can affect hatching time, how were these parameters controlled? As it is, it is not possible to disentangle room effects from treatment effects.
In several oviparous species, like fish, the spectral composition of light has an influence on embryonic development. The spectral composition and intensity of the light has to be mentioned. Was this type of light used in previous experiments and was it controlled for effects on embryonic development? What was the distance between the light and the embryos? Was it a source of extra heat which could have an influence on humidity? Was this distance equivalent between light and photoperiod treatments.
Lines 183-190: this part is unclear. For the photoperiod treatment, were the bins wrapped and unwrapped daily? In that case, did this group received extra manipulations compared to the others? The authors mentioned line 101 that undisturbed eggs hatched after 6 or 7 days. Did the present embryos hatched earlier than expected? In Salica et al., 2017, it is stated that a stress like dehydration engenders earlier and more synchronous hatching. Could the authors rule out the hypothesis that differences in vibrations and/or humidity, temperature and manipulations may have affect the results?
Lines 192-251: This part needs to be revised. The relevance of observing whole-clutch or individual-egg flooding needs to be scientifically explained with clear hypotheses.
I did not get why the authors gave details regarding embryos tested in 2014. If I understood well, these embryos are not included in the present paper. In addition, the method is not exactly the same than for 2015. This is very confusing and should be removed.
Lines 215-220: very few details are given regarding flooding of whole clutches. As it is, it is not possible to replicate and to understand what was done. How were light-cue orientation tested on clutches?
Lines 222-251: A schematic representation should be given to understand the protocol. If I understand well, the protocol was changed during the experiment (wrapping with black tape (line 232) and change in the light distance (line 245)). Why? Was the exposure to light not controlled enough during the first part? The authors have to test more properly the potential effects of these factors before conducting statistics on pooled data. The authors present two experiments related to these changes (e.g. fig 7) but I would bot called this “experiment” since nothing is tested.
As a control for side bias (or laterality), the authors used 20 additional embryos which were not exposed to light. In my opinion, this is not a good control because of environmental differences. All tested embryos were exposed to light and they may have used environmental cues present in the environment to orient their behaviour. The authors have to mention if the sides “light” or “dark” were counterbalanced between the right and the left between trials. And, the occurrence of “hatching to the right” or “hatching to the left” has to tested against chance level.
The intensity of the light should be mentioned. Was it comparable to what is found in natural conditions?

Validity of the findings

Statistical analysis: lines 255-277: there is a mix between parametric and non-parametric statistics. The authors should clearly state if the requirements to use parametric statistics were met.
Lines 259: the sentence is not clear. Does it mean proportions? Does per “clutch” refers to the split-clutch (7 or 10 eggs)? I would recommend the authors to present data as box plots with the different time-points or specific points like the time points presented Fig 3 (4.5, 5, 5.5, 6; 6.5 and 7 days of age). Hatching success should also be mentioned. The authors focused on specific time points around 5.75, 5.67, 5.83 days but it does not provide a more general view of the process (onset of hatching, 50% hatching, hatching completion). Hatching synchrony must be defined more properly. How was the criteria “maximum proportion” defined? Why a specific 2h window (line 268-269) and not 1 or 3 hour-window? What is commonly observed in undisturbed embryos under natural conditions?
Line 271: what is meant by “compared the distribution of these modal hatching times across treatment”. This is unclear.
I would recommend the authors to provide first, a descriptive view of the course of hatching in each treatment (e.g. time of onset, 50%...). And secondly, to provide comparisons between groups.
Lines 280-281: The authors mentioned: “ based on the null hypothesis that the probability of hatching into open water by chance was equal to the natural proportion of exposed surface area of embryos”. I did not get the meaning. The same for the hypothesis of “the proportion of hatching into the glass in individual trials was equal to the observed average incidence of embryos hatching into the jelly in whole clutch..”
I did not get these hypotheses. If the authors expected embryos to hatched towards the light, then they should just focus on the comparisons of the proportion of embryos hatched towards the light compared to chance level 50%. Laterality bias should be tested properly.
Why the authors expected an effect of the side of insertion in the tube? Does it influence the orientation of the head of the embryo at the start of the experiment?
Lines 334-340: the authors mentioned a high proportion of hatching “failure” in individual embryos. Were the tubes used to maintain embryos previously used in the lab? How is this explained?
The discussion section needs to be revised according to the points raised. In addition the authors provide very general information which are not based on their data.
The discussion is speculative and not supported by data. For example, there is only once sentence lines “449-450” related to the results of the experiment on hatching direction. The authors state that the embryos use light cues but this is not demonstrated. The authors should discuss the parameters observed. They should also discuss the limitation of their study. Validity and accuracy of the data should be discussed. The hypotheses mentioned in the introduction section should be discussed in light of the results. The present data should also be compared to data obtained previously in the lab.

---

## Round 0.2 · Minor Revisions

I really appreciate the work done with your manuscript. Thank you for addressing all the reviewers comments in your revised version. However, I agree with the last suggestions of our reviewer, and think that you need to balance your methodological details with reading fluency. Would you please revise your manuscript having in mind these new comments?

Reviewer 1 ·

Basic reporting

After reading carefully the suggestions proposed by Reviewer 2 and myself I found that authors were very open to the revision and I celebrate that they responded with extreme detail to all the questions and suggestions expressed by the reviewers.

However, in my opinion, with the intention to reinforce and explain in detail why and when some of the methods were employed Discussion resulted too long. Particularly, those sections that identify caveats to the major findings and interpretations of the results (Discussion, “data interpretation and limitations of experiment” and “Hatching orientation – data interpretation, evidence, and limitations”), then it could be abbreviated and or moved to Materials and Methods.

Experimental design

Experimental design was improved, by including photographs and figures as was suggested. Also, I suggest to move information that was included in Discussion to improve the experimental methodology

Validity of the findings

This manuscript provide novel data about the rol of light shifting hatching timing and orientation in an amphibian on undisturbed conditions. Methods resulted adequate to the proposed questions; statistics is correct and in this new version authors worked hard to explain experimental design in detail. I consider that Discussion needs some changes. Conclusions are appropriate and conneted to the obseved results.

Additional comments

I think authors did the best to improve the manuscript, I appreciate that; however, in my opinion, some of the information included in the new version of the manuscript needs to be worked.

I would recommend to the authors:

Section: Diel pattern of hatching – data interpretation and limitations of experiment
1) To move this entire paragraph to M&M:
“A limitation of our experiment, imposed by field-season logistics, is that we only had one light-controlled compartment per treatment; thus, any other environmental variation between compartments was confounded with light treatment. In general, variation in temperature and humidity can influence the hatching timing of amphibians via effects on development rate (Duellman & Trueb 1986; Seymour & Bradford 1995). In A. callidryas these variables may also directly affect embryos’ hatching decisions; both drying and heating can induce hatching (Salica et al. 2017). For instance, during the warm El Niño conditions in 2015, embryos in our laboratory developed faster and hatched earlier than in other years (see Warkentin et al. 2017). All treatments were run simultaneously, on split-clutches, and eggs in different compartments were separated by only a short distance and thin plastic sheet. Thus they were subjected concurrently to any overall temperature fluctuations in the room. We used LED lights to minimize heat from the bulbs and manual misting to maintain humidity. In our few spot checks, all compartments were within a range of 0.6°C, but we do not have continuous temperature data through the experiment.”

2) To delete the following paragraph, this information does not contribute with testable or comparable data in the discussion.
“Furthermore, these experimental results are consistent with unquantified prior observations of A. callidryas in many different contexts; continuous light generally appears to delay hatching and hatching often peaks shortly after natural dusk or experimental lights-out (Warkentin, personal observations”.

Section: Hatching orientation – data interpretation, evidence, and limitations
I suggest to move to Materials and Methods the following paragraphs:
1) “Their poorer orientation in glass cups suggests that any potential memory of their prior oxygen gradient is insufficient as a cue, and that visual cues may be relevant. Our phototaxis experiment was designed to test for a contribution of light or visual cues “.

2) “The way that we constructed the cups precluded an equivalent “both sides light” control to test for side-bias due to aspects of the visual environment. Thus, our experiment does not address whether embryos might tend to orient hatching toward or away from any particular visual stimulus, simply whether they orient toward light and visual cues rather than their absence. We did not measure light intensity in this experiment, nor do we have field measurements of underwater light intensity in A. callidryas ponds”.

In general I am pleased with the changes made on this new version of the Manuscript, but Discussion needs to be shorten and I suggest that some of the paragraphs could be moved to Materials and methods.

---

## Round 0.3 · accepted · Accept

I am happy to inform you that your manuscript is ready for publication. Nice work!

#